# Multicenter Diagnostic Evaluation of OnSite COVID-19 Rapid Test (CTK Biotech) among Symptomatic Individuals in Brazil and the United Kingdom

Caitlin R. Thompson,[a] Pablo Muñoz Torres,[b] Konstantina Kontogianni,[a] Rachel L. Byrne,[a] LSTM Diagnostic group, Saidy Vásconez Noguera,[b,c] Alessandra Luna-Muschi,[b,c] Ana Paula Marchi,[b,c] Pâmela S. Andrade,[d] Antonio dos Santos Barboza,[e] Marli Nishikawara,[e] CONDOR steering group, Richard Body,[h] Margaretha de Vos,[g] Camille Escadafal,[f] Emily Adams,[a,i] Silvia Figueiredo Costa,[b,c] Ana I. Cubas-Atienzar[a]

[a]Liverpool School of Tropical Medicine, Centre for Drugs and Diagnostics, Liverpool, United Kingdom
[b]LIM-49, Instituto de Medicina Tropical, Faculdade de Medicina da Universidade de São Paulo, São Paulo, Brazil
[c]Departamento de Moléstias Infecciosas e Parasitárias, Faculdade de Medicina da Universidade de São Paulo, São Paulo, Brazil
[d]Department of Epidemiology, School of Public Health of University of São Paulo, São Paulo, Brazil
[e]Centro de atendimento ao colaborador, Hospital das Clínicas da Faculdade de Medicina da Universidade de São Paulo, São Paulo, Brazil
[f]Divisão de Laboratório Central, Hospital das Clinicas, Faculdade de Medicina da Universidade de São Paulo, São Paulo, Brazil
[g]FIND, Geneva, Switzerland
[h]Manchester University NHS Foundation Trust, Manchester, United Kingdom
[i]Global Access Diagnostics, Thurleigh, Bedfordshire, United Kingdom

Silvia Figueiredo Costa and Ana I. Cubas Atienzar are joint last senior authors.

**ABSTRACT** The COVID-19 pandemic has given rise to numerous commercially available antigen rapid diagnostic tests (Ag-RDTs). To generate and to share accurate and independent data with the global community requires multisite prospective diagnostic evaluations of Ag-RDTs. This report describes the clinical evaluation of the OnSite COVID-19 rapid test (CTK Biotech, CA, USA) in Brazil and the United Kingdom. A total of 496 paired nasopharyngeal (NP) swabs were collected from symptomatic health care workers at Hospital das Clínicas in São Paulo, Brazil, and 211 NP swabs were collected from symptomatic participants at a COVID-19 drive-through testing site in Liverpool, United Kingdom. Swabs were analyzed by Ag-RDT, and results were compared to quantitative reverse transcriptase PCR (RT-qPCR). The clinical sensitivity of the OnSite COVID-19 rapid test in Brazil was 90.3% (95% confidence interval [CI], 75.1 to 96.7%) and in the United Kingdom was 75.3% (95% CI, 64.6 to 83.6%). The clinical specificity in Brazil was 99.4% (95% CI, 98.1 to 99.8%) and in the United Kingdom was 95.5% (95% CI, 90.6 to 97.9%). Concurrently, analytical evaluation of the Ag-RDT was assessed using direct culture supernatant of SARS-CoV-2 strains from wild-type (WT), Alpha, Delta, Gamma, and Omicron lineages. This study provides comparative performance of an Ag-RDT across two different settings, geographical areas, and populations. Overall, the OnSite Ag-RDT demonstrated a lower clinical sensitivity than claimed by the manufacturer. The sensitivity and specificity from the Brazil study fulfilled the performance criteria determined by the World Health Organization, but the performance obtained from the UK study failed to do. Further evaluation of Ag-RDTs should include harmonized protocols between laboratories to facilitate comparison between settings.

**IMPORTANCE** Evaluating rapid diagnostic tests in diverse populations is essential to improving diagnostic responses as it gives an indication of the accuracy in real-world scenarios. In the case of rapid diagnostic testing within this pandemic, lateral flow tests that meet the minimum requirements for sensitivity and specificity can play a key role in increasing testing capacity, allowing timely clinical management of those

Address correspondence to Silvia Figueiredo Costa, silviacosta@usp.br, or Ana I. Cubas-Atienzar, Ana.CubasAtienzar@lstmed.ac.uk.

The authors declare a conflict of interest. E.A. is an employee of Mologic. C.E. and M.D.V. are employees of FIND. E.A., C.E., and M.D.V. had no role in data collection and analysis. The other authors have no conflicts to declare.

infected, and protecting health care systems. This is particularly valuable in settings where access to the test gold standard is often restricted.

**KEYWORDS** COVID-19, RDT, diagnostics

To meet the immense diagnostic demand of the COVID-19 pandemic, the use of rapid diagnostic tests for the detection of SARS-CoV-2 antigens (Ag-RDTs) has become a priority. To date, there are currently 321 SARS-CoV-2 Ag-RDTs on the market or in development according to the Foundation for New Innovative Diagnostics (FIND) (accessed March 2022) (1). However, clinical evaluation of these Ag-RDTs has been relatively limited, and performance results differ greatly between studies (2, 3). In the United Kingdom, the use of Ag-RDTs has been integral to reducing the spread of COVID-19 (4). However, since April 2022 the UK government has ceased free Ag-RDT testing, now requiring the responsibility of the purchase and use of the test to be placed on the individual.

In Brazil, the national SARS-CoV-2 testing approach has been insufficient in its use of this Ag-RDTS as a diagnostic tool in the efforts to contain this pandemic (5). Many initiatives such as recruiting capacity in university research laboratories and biotechnological enterprises, investments in new laboratory infrastructure, and fast-track regulatory measures were launched to scale up SARS-CoV-2 quantitative reverse transcriptase PCR (RT-qPCR) testing in Brazil. However, RT-qPCR capacity has not been sufficient to control the progress of the pandemic (5).

Despite the commercialization of several vaccines for SARS-CoV-2, the COVID-19 pandemic is still ongoing due to vaccine inequity (6), uneven vaccine uptake between populations (7), and the emergence of new highly transmissible variants of SARS-CoV-2 (8).

The gold standard for diagnosis of COVID-19 remains the detection of SARS-CoV-2 RNA. However, RT-qPCR requires skilled laboratory scientists, installed capacity, and expensive consumables and reagents, which can be challenging to implement in low- and middle-income countries (LMIC), where the burden of COVID-19 is disproportionately felt. Additionally, turnaround of results of RT-qPCR can take up to 1 week (9).

In order to continue to meet the challenges of testing capacity, prospective diagnostic evaluation studies across multiple, independent sites are required to determine the accuracy of COVID-19 Ag-RDTs available for purchase by the public.

In this study, the OnSite COVID-19 rapid test (CTK Biotech) was evaluated against the SARS-CoV-2 diagnostic gold standard RT-qPCR. Testing was undertaken in Brazil and the United Kingdom across different settings: on health care workers (HCWs) at Hospital das Clínicas, a tertiary-care hospital affiliated with the University of São Paulo (Brazil), and at a National Health Service (NHS) COVID-19 drive-through community testing center in Liverpool, United Kingdom.

## RESULTS

**Clinical evaluation.** The demographics of both the Brazilian and UK study cohorts are shown in Table 1. In Brazil the median number of days from onset of symptoms was 3 (Q1 (lower quartile) to Q3 (higher quartile), 2 to 4), with a vaccination rate of 96.5% (including partially and fully vaccinated participants). In the United Kingdom, the median number of days from symptom onset was 2 (Q1 to Q3, 1 to 3) and the vaccination rate was 84.9% (including partially and fully vaccinated participants). Significantly higher SARS-CoV-2 RT-qPCR positivity was detected in the United Kingdom (36.5%; 95% confidence interval [CI], 0.29 to 0.43) than in Brazil (6.5%; 95% CI, 0.05 to 0.09) ($P < 0.05$).

The clinical sensitivity of the Onsite Ag-RDT across evaluation sites was heterogeneous, with a clinical sensitivity of 90.3% (95% CI, 75.1 to 96.7%) in Brazil and 75.3% (95% CI, 64.6 to 83.6%) in the United Kingdom (Table 2). The difference in sensitivities between sites was not statistically significant ($P = 0.128$). The clinical specificity of the Onsite Ag-RDT was 99.4% (95% CI, 98.1 to 99.8%) in Brazil and 95.5% (95% CI, 90.6 to 97.9%) in the United Kingdom.

In Brazil, of the 496 participants included, 32 were SARS-CoV-2 RT-qPCR positive (6.5%) (Table 2). Twenty-eight of the RT-qPCR-positive samples (90.3%) were Ag-RDT positive, while 3 (9.7%) were Ag-RDT negative and one was invalid (3.1%). Invalid results were

**TABLE 1** Demographics of Ag-RDT clinical evaluation cohorts for Brazil and the United Kingdom

| | Value for country: | |
|---|---|---|
| Category | Brazil | United Kingdom |
| Age, yr [mean (minimum–maximum), N] | 38.1 (16–69), 496 | 40.8 (20–86), 211 |
| Gender [% F[b] (n/N)] | 71.5% (354/495)[a] | 52.4% (110/210)[a] |
| Symptoms present [% yes (n/N)] | 99.6% (494/496) | 100% (211/211) |
| Days from symptom onset [median (Q1–Q3), N] | 3 (2–4), 494 | 2 (1–3), 211 |
| Days 0–3 (n, %) | 294, 59.3% | 169, 80.1% |
| Days 4–7 (n, %) | 186, 37.5% | 36, 17.1% |
| Days 8+ (n, %) | 14, 2.8% | 6, 2.8% |
| Vaccinated (n, %) | 460, 92.7% | 132, 62.6% |
| Partially vaccinated (n, %) | 19, 3.8% | 47, 22.3% |
| Not vaccinated (n, %) | 10, 2.0% | 32, 15.2% |
| Vaccination not disclosed (n, %) | 7, 1.4% | 1, 0.5% |
| SARS-CoV-2 positivity [% (n/N)] | 6.5% (32/496) | 36.5% (77/211) |

[a]Gender was not disclosed for two participants.
[b]F, female.

removed for further analysis. Of the 464 RT-qPCR-negative samples, 3 were Ag-RDT positive (0.6%). The sensitivity and specificity of the OnSite Ag-RDT on RT-qPCR were 90.3% (95% CI, 75.1 to 96.7%) and 99.4% (95% CI, 98.1 to 99.8%), respectively (Table 2). Sensitivity for ≤7 days since symptom onset was 96.2% (95% CI, 81.1 to 99.3%). Sensitivity according to cycle threshold ($C_T$) value was 95.0% (95% CI, 75.1 to 99.8%) for a $C_T$ value of ≤25 and 90.3% (95% CI, 75.1 to 96.7%) for a $C_T$ value of ≤33 (Table 3). No statistically significant difference was found in sensitivity between different $C_T$ value groups.

In the United Kingdom, of the 211 participants recruited, 77 (36.5%) were SARS-CoV-2 RT-qPCR positive (Table 2). Fifty-eight (75.3%) of the 77 RT-qPCR-positive samples were also Ag-RDT positive, while 19 (24.7%) were Ag-RDT negative. Of the 134 RT-qPCR-negative samples, 128 (95.5%) were also Ag-RDT negative and 6 (4.5%) were Ag-RDT positive. For the UK evaluation, the sensitivity and specificity were 75.3% (95% CI, 64.6 to 83.6%) and 95.5% (95% CI, 90.6 to 97.9%), respectively. Sensitivity for ≤7 days since symptom onset was 76.7% (95% CI, 65.8 to 84.9%). $C_T$ values of ≤20, ≤25, ≤33, and ≤40 had a sensitivity of 90.5% (95% CI, 77.4 to 97.3%), 80.3% (95% CI, 69.2 to 88.1%), 76.3% (95% CI, 65.5 to 84.5%), and 75.3% (95% CI, 64.6 to 83.6%), respectively. Sensitivity was statistically higher among samples with $C_T$ values of ≤20 compared with samples with $C_T$ values of ≤33 ($P = 0.029$) and ≤40 ($P = 0.044$).

Subgroup analyses of the Brazilian and UK evaluation cohorts (Table 4) were performed to determine any associated differences in sensitivity compared to vaccination status and days from symptom onset. In the Brazilian cohort, the sensitivity of the OnSite Ag-RDT was significantly lower on samples from patients with >7 days since symptom onset compared to samples with 0 to 3 days since symptom onset ($P = 0.02924$) and samples with 0 to 7 days of onset ($P = 0.03115$), but no differences in sensitivity were found between groups of different vaccination statuses. In the UK, no difference in sensitivity was observed between groups of different days since symptom onset and vaccination status (all P values

**TABLE 2** Results and clinical sensitivity and specificity of the OnSite COVID-19 Ag device based on COVID-19 RT-qPCR results in Brazil and the United Kingdom[a]

| Result of OnSite COVID-19 Ag device | No. confirmed by RT-qPCR in country: | | | | | |
|---|---|---|---|---|---|---|
| | Brazil[b] | | | United Kingdom[c] | | |
| | Positive | Negative | Total | Positive | Negative | Total |
| Positive | 28 | 3 | 31 | 58 | 6 | 64 |
| Negative | 3 | 461 | 464 | 19 | 128 | 147 |
| Total | 31 | 464 | 495 | 77 | 134 | 211 |

[a]RT-qPCR, real-time quantitative reverse transcriptase PCR; $C_T$, cycle threshold; CI, confidence interval.
[b]For results from Brazil, clinical sensitivity was 90.3% (95% CI, 75.1 to 96.7%; N = 31), clinical specificity was 99.4% (95% CI, 98.1 to 99.8%, N = 464), and the invalid rate was 0.2% (n/N = 1/496).
[c]For results from the United Kingdom, clinical sensitivity was 75.3% (95% CI, 64.6 to 83.6%, N = 77), clinical specificity was 95.5% (95% CI, 90.6 to 97.9%, N = 134), and the invalid rate was 0% (n/N = 211/211).

**TABLE 3** COVID-19 RT-qPCR results in Brazil and the United Kingdom

| | Value for country: | |
|---|---|---|
| Category | Brazil | United Kingdom |
| PCR $C_T$ [median (Q1–Q3); $N$] | 19.6 (17.52–23), 31 | 19.5 (17.3–22.8), 77 |
| $C_T$ ($n$, %) | | |
| >33 | 0, 0% | 1, 1.3% |
| >30 | 1, 3.2% | 5, 6.5% |
| >25 | 7, 22.6% | 11, 14.3% |
| Sensitivity by $C_T$ (95% CI), $N$ | | |
| ≤20 | 100.0% (76.8–100%), 14 | 90.5% (77.4–97.3%), 42 |
| ≤25 | 95.0% (75.1–99.8%), 20 | 80.3% (69.2–88.1%), 66 |
| ≤33 | 90.3% (75.1–96.7%), 31 | 76.3% (65.5–84.5%), 76 |
| ≤40 | NA[a] | 75.3% (64.6–83.6%), 77 |

[a]NA, not available; maximum RT-qPCR cutoff was ≤33 in Brazil.

of >0.05). In Brazil, 52% of the positive samples were classified as Delta and 39% as Gamma. In the United Kingdom, variant determination was not performed, but at the time of enrollment, 100% of genome submissions corresponded to the Delta variant (10).

**Analytical sensitivity.** The limit of detection (LOD) of the OnSite Ag-RDT was $1.0 \times 10^3$ PFU/mL, $1.0 \times 10^3$ PFU/mL, $1.0 \times 10^2$ PFU/mL, $5.0 \times 10^3$ PFU/mL, and $1.0 \times 10^3$ PFU/mL when tested on the wild-type (WT), Alpha, Delta, Gamma, and Omicron lineages, respectively. This gave a viral copy equivalent of approximately $2.1 \times 10^5$ copies/mL, $2.1 \times 10^4$ copies/mL, $1.6 \times 10^4$ copies/mL, $3.5 \times 10^6$ copies/mL, and $8.7 \times 10^4$ copies/mL for the Ag-RDT for the WT, Alpha, Delta, Gamma, and Omicron lineages, respectively.

## DISCUSSION

The study aimed to evaluate the diagnostic performance of the OnSite COVID-19 Ag rapid test (CTK Biotech) in two different settings. Evaluating rapid diagnostic tests in diverse populations is vital to improving diagnostic responses as it gives an indication of the diagnostic accuracy in real-world scenarios. In the case of rapid diagnostic testing within this pandemic, lateral flow tests which meet the minimum requirements for sensitivity and specificity can play a key role in increasing testing capacity, allowing timely clinical management of those infected and protecting health care systems (11). This is particularly valuable in settings where access to the gold standard RT-qPCR is often not available. Ag-RDTs are low cost, are easy to use, and do not require specialized skills or equipment, which is essential to promote universal access.

The sensitivity and specificity of the OnSite Ag-RDT in a hospital setting in Brazil fulfilled the performance criteria determined by the World Health Organization (WHO). However, the

**TABLE 4** Ag-RDT result by onset of symptoms and vaccinated individuals in Brazil and the United Kingdom[c]

| | Brazil | | | | United Kingdom | | | |
|---|---|---|---|---|---|---|---|---|
| Category | Ag-RDT positive ($n$, %) | Ag-RDT negative ($n$, %) | Sensitivity, %[a] | 95% CI | Ag-RDT positive ($n$, %) | Ag-RDT negative ($n$, %) | Sensitivity, %[a] | 95% CI |
| Days from symptom onset | | | | | | | | |
| 0–3 | 16, 5.4% | 278, 94.6% | 100.0% | 76.9–100.0% | 52, 30.6% | 117, 69.4% | 79.7% | 67.2–89.0% |
| 4–7 | 12, 6.4% | 173, 93.6% | 91.7% | 61.5–99.8% | 10, 27.8% | 26, 72.2% | 64.3% | 35.2–87.3% |
| 8+ | 3, 21.4% | 11, 78.6% | 60.0% | 14.7–94.7% | 2, 33.3% | 4, 66.7% | 66.7% | 9.4–99.2% |
| Vaccination received | | | | | | | | |
| Vaccinated[b] | 31, 6.5% | 447, 93.5% | 93.3% | 77.4–99.2% | 52, 29.3% | 126, 70.7% | 78.3% | 65.8–87.9% |
| Not vaccinated | 1, 10.0% | 9, 90.0% | 0% | NA | 11, 34.4% | 21, 65.6% | 62.5% | 35.4–84.8% |
| Not disclosed | 0, 0% | 7, 100.0% | NA | NA | 1, 100.0% | 0, 0% | 100% | 2.5–100.0% |

[a]Compared to RT-qPCR.
[b]Vaccinated defined as 1 or more doses.
[c]RT-qPCR, real-time quantitative reverse transcriptase PCR; CI, confidence interval; NA, not available.

sensitivity obtained in a community setting at a drive-through testing site in the United Kingdom missed the minimum recommendations (12) for both sensitivity and specificity. In guidance published by the WHO, minimum performance requirements for an Ag-RDT include a sensitivity of >80% and specificity of >97% (12). Analytical evaluation of the OnSite Ag-RDT detected wild-type, Alpha, Delta, and Omicron viruses, meeting the recommendations in the WHO target product profile for SARS-CoV-2 Ag-RDT of an acceptable analytical LOD of $1.0 \times 10^6$ RNA copies/mL (13) with the Gamma variant slightly outside this threshold. In the Brazilian cohort, the Gamma variant was responsible for 39% of infections and the Delta variant was responsible for 52%. This is an interesting finding as it does not reflect the wider variant circulation in Brazil during this period as the Gamma variant was responsible for over 93% of infections in July 2021 and 70% of infections in August 2021 followed by Delta at 5%, rising to 29%, respectively (14). In the United Kingdom, positive RT-qPCR results were not sequenced, but it is assumed that all infections were Delta (B.1.617.2) due to the >99% circulation of this variant in the United Kingdom during the time of collection (15).

In both settings, the Ag-RDT had a higher sensitivity in samples with lower RT-qPCR cycle threshold ($C_T$) values; this is consistent with other Ag-RDT studies (16). The sensitivity of the Ag-RDT was also highest when time since symptom onset was 3 days, decreasing between 4 and 7 days and again after 8 days since symptom onset in both settings. Interestingly, in the United Kingdom cohort, the sensitivity slightly increased between 4 and 7 days and 8+ days, from 64.3% to 66.7%. However, the sample size of the 8+-day group was too small to be statistically significant, and therefore, a larger sample set would be needed to provide significance. In Brazil, 93.3% of the cohort was vaccinated due to the vaccination efforts of the country and the prioritization of health care workers in the vaccination program (17). In the United Kingdom, there was a larger number of nonvaccinated people; however, the differences in vaccinated and nonvaccinated people were not statistically significant for either cohort. A larger sample set would have to be used, and further analysis of these subgroups would have to take place, in order to provide any significant data.

This study has several strengths: it is a multicenter and multinational evaluation across two different settings with differing testing capacities, prevalences of SARS-CoV-2, and population characteristics. In Brazil, samples were taken from a very exclusive population, health care workers in a health care setting with a high vaccination uptake compared to the rest of the population (18). In the United Kingdom, data were collected from a diverse population, any person over the age of 18 presenting with COVID-19 symptoms at a government-run, drive-through COVID-19 testing facility. It is important to evaluate Ag-RDTs in a heterogeneous population and setting to obtain meaningful diagnostic accuracy data.

The main limitation for the study is that the drive-through testing setting in the United Kingdom did not allow for Ag-RDT testing to be performed at the point of care just after sample collection as recommended by the instructions for use (IFU). Guidance in the United Kingdom restricted testing of suspected COVID-19-positive individuals to high-containment laboratories. Currently, there are limited studies on the stability of Ag-RDTs. A systematic review on Ag-RDTs did not find a significant difference between 96 data sets that involved fresh specimens for antigen testing and 23 data sets including freeze-thawed specimens for antigen testing (19). Although, it is not stated whether the swabs were freeze-dried or used with transport buffer. However, one review of Ag-RDT performance in sub-Saharan Africa suggested that a delay in performing the test (Coris COVID-19 Ag Respi-strip) may impact its stability if samples are stored at 4℃ rather than frozen at −20℃ immediately (20). Conversely, studies have shown that SARS-CoV-2 RNA remains stable for up to 9 days in dry swabs at an ambient temperature of 20℃ (21), and proteins have been shown to be more stable than RNA (22). Therefore, further investigation must take place to determine whether time from sample collection to Ag-RDT testing has a significant impact on the sensitivity.

Two other limitations of this study are that the RT-qPCR methodologies varied between the two cohorts and that there were differences in SARS-CoV-2 prevalence. These factors have been described as a major cause of index case diagnostic accuracy (23). For future evaluations, quantification of the viral copy numbers rather than $C_T$ values is recommended to mitigate differences in RT-qPCR assay performances. This $C_T$

variability has been estimated to be >1,000-fold in viral copy numbers per milliliter (23), as the RT-qPCR used in the United Kingdom has an LOD 10-fold more sensitive (10 genome copies/mL) than that of the RT-qPCR used in Brazil (100 genome copies/ mL) (24). The higher sensitivity of the RT-qPCR assay used in the United Kingdom, together with the higher cutoff used ($C_T$ of 40 versus $C_T$ of 32 to 33 in Brazil), could have contributed to higher numbers of false negatives in the index test than in the Brazilian cohort. Additionally, there is a significant difference in sample size and in confirmed RT-qPCR positives (SARS-CoV-2 prevalence) between the two cohorts, with a low number of positive samples found in the Brazilian evaluation (6.3%) compared to the United Kingdom evaluation (36.5%). It has been reported that differences in prevalence can have an effect on the sensitivity and specificity of index tests (25, 26).

In conclusion, the data indicate that the OnSite Ag-RDT had lower performance quality than that published by the manufacturers for the detection of SARS-CoV-2 in clinical samples and varied greatly between the two settings in this study. Further evaluation of the use of Ag-RDTs should strictly follow the IFU of the test and include harmonized protocols between laboratories to facilitate comparison between settings. In particular, the use of viral copy numbers rather than $C_T$ values has been suggested to minimize the variability between laboratories.

## MATERIALS AND METHODS

**Clinical evaluation.** This was a prospective evaluation of consecutive participants enrolled in two different settings.

**(i) Brazil.** Health care workers (HCWs) with suspected COVID-19 symptoms (fever, cough, shortness of breath, tight chest, runny nose, sore throat, anosmia, ageusia, headache, and diarrhea) were enrolled at the HCW service of Hospital das Clínicas in São Paulo from July to October 2021. Ethical approval was obtained from the Hospital's Ethics Committee with the CAAE number 35246720.0.0000.0068. Informed consent was obtained from all study participants for respiratory samples and clinical data collection.

Participants were clinically evaluated, and RT-qPCR for SARS-CoV-2 was performed from combined nasopharyngeal (NP) and oropharyngeal swabs (Goodwood Medical Care LTD/[DG] China) following the national standard of care. Following the RT-qPCR swabs, nasopharyngeal (NP) swabs were collected for Ag-RDT testing. The OnSite Ag-RDT was performed at the point of care by HCWs following the manufacturer's instructions for use (IFU).

For SARS-CoV-2 RT-qPCR, RNA was extracted from 0.9% saline solution with an automated method using magnetic beads (RNA sample preparation system; Abbott, IL, USA). SARS-CoV-2 RT-qPCR was performed using an adapted protocol described by Corman et al. (27) to detect the E gene as the first-line screening tool, followed by confirmatory testing with an assay detecting the N gene (Abbott, USA) and the commercial SARS-CoV-2 N1+N2 RT-qPCR kit to detect N1 and N2 genes (Qiagen, USA). A SARS-CoV-2 RT-qPCR result was considered positive with an amplification cycle threshold ($C_T$) value of ≤32 and a $C_T$ value of ≤33, respectively.

For the detection of SARS-CoV-2 variants, samples were amplified using the TaqPath one-step RT-qPCR master SARS-CoV-2 mutation panel assay (40×) (Thermo Fisher Scientific, Waltham, MA, USA) following the manufacturer's instructions. The RT-qPCR mixture was prepared, and samples were tested for the presence of each S-gene mutation. The mutation panel was customized to detect each variant as follows: Alpha (P681H[+], E484K[−], K417N[−], L452R[−], T20N[−], P681R[−], L452Q[−]), Beta (E484K[+], K417N[+], P681H[−], L452R[−], T20N[−], P681R[−], L452Q[−]), Gamma (E484K[+], T20N[+], K417N[−], L452R[−], P681H[−], P681R[−], L452Q[−]), Delta and Kappa (L452R[+], P681R[+], E484K[−], K417N[−], T20N[−], P681H[−], L452Q[−]), Zeta (E484K[+], K417N[−], L452R[−], T20N[−], P681H[−], P681R[−], L452Q[−]), and Lambda (L452Q[+], E484K[−], K417N[−], P681H[−], L452R[−], T20N[−], P681R[−]) (28–30). Data were analyzed by QuantStudio design and analysis software v2.5.1. in the genotyping module. The variant was identified according to the positivity for each mutation tested.

**(ii) United Kingdom.** In the United Kingdom, adults presenting with symptoms of COVID-19 (fever, cough, shortness of breath, tight chest, runny nose, sore throat, anosmia, ageusia, headache, diarrhea, and tiredness) at a national community testing facility, the Liverpool John Lennon Airport drive-through COVID-19 test center, were asked to participate in the study. Participants were recruited between July and August of 2021 under the Facilitating Accelerated COVID-19 Diagnostics (FALCON) study. Ethical approval was obtained from the National Research Ethics Service and the Health Research Authority (IRAS identifier [ID], 28422; clinical trial ID, NCT04408170).

Swabs were taken systematically; first an NP swab sample in universal transport medium (UTM) was collected from the patient for the reference RT-qPCR test, and then an NP swab sample was taken to perform the Ag-RDTs. Due to biosafety restrictions at the drive-through center, Ag-RDT testing was not done immediately after sample collection as per the IFU. All samples were transported in insulated UN3373 transit bags to the Liverpool School of Tropical Medicine (LSTM) and processed upon arrival by trained laboratory researchers following the IFU. Processing happened within a maximum of 3 h of collection. Ag-RDTs were performed, and the UTM NP swab samples were aliquoted and stored at −80°C until RNA extraction. RNA was extracted using the QIAamp 96 Virus QIAcube HT kit (Qiagen, Germany)

on the QIAcube (Qiagen, Germany) and screened using TaqPath COVID-19 (ThermoFisher, United Kingdom) on the QuantStudio 5 thermocycler (ThermoFisher, United Kingdom). The SARS-CoV-2 RT-qPCR result was considered positive if any two of the three targets (N, ORFab, and S) were amplified with a cycle threshold ($C_T$) value of $\leq$40.

**Analytical sensitivity (United Kingdom only).** Viral culture methods to propagate SARS-CoV-2 isolates and to calculate PFU per milliliter followed those previously described (31). Briefly, isolates of SARS-CoV-2 from the wild-type (Pango, B1) (REMRQ0001/Human/2020/Liverpool, GISAID ID EPI_ISL_464183), Alpha (B.1.1.7) (SARS-CoV-2/human/GBR/FASTER_272/2021, GenBank ID MW980115), Delta (B.1.617.2) (SARS-CoV-2/human/GBR/Liv_273/2021, GenBank ID OK392641), Gamma (P.1) (hCoV-19/Japan/TY7-503/2021, GISAID ID EPI_ISL_792683), and Omicron (BA.1) (SARS-CoV-2/human/GBR/Liv_1326/2021, Genebank ID OP630952) lineages were used to evaluate the limit of detection (LOD) of the OnSite Ag-RDT. For the determination of the LOD, a fresh aliquot was serially diluted from $1.0 \times 10^5$ PFU/mL to $1.0 \times 10^2$ PFU/mL. Each dilution was tested in triplicate. Twofold dilutions were made below the 10-fold LOD dilution to confirm the lowest LOD (LLOD).

Viral RNA was extracted from each dilution using the QIAamp viral RNA minikit (Qiagen, Germany) according to the manufacturer's instructions and quantified using Genesig RT-qPCR (Primer Design, United Kingdom). Genome copy number (gcn) per milliliter was calculated as previously described (32).

**Statistical analysis.** The sensitivity and specificity, with 95% confidence intervals (CIs), were calculated based on the results of the reference method by RT-qPCR assay. Statistical analyses were performed using R scripts, Epi Info, and GraphPad Prism 9.1.0 (GraphPad Software, Inc., CA). The 95% confidence interval (CI) for the sensitivity and specificity was calculated using Wilson's method. Two-tailed Fisher's exact and chi-square tests were used to determine nonrandom associations between categorical variables. Statistical significance was set at $<$0.05.

## ACKNOWLEDGMENTS

LSTM diagnostic group: Caitlin Greenland Bews, Kate Buist, Karina Clerkin, Thomas Edwards, Lorna Finch, Helen R. Savage, Jahanara Wardale, Rachel Watkins, Chris Williams, Dominic Wooding.

CONDOR steering group: A. Joy Allen, Julian Braybrook, Peter Buckle, Eloise Cook, Paul Dark, Kerrie Davis, Gail Hayward, Adam Gordon, Anna Halstead, Charlotte Harden, Colette Inkson, Naoko Jones, William Jones, Dan Lasserson, Joseph Lee, Clare Lendrem, Andrew Lewington, Mary Logan, Massimo Micocci, Brian Nicholson, Rafael Perera-Salazar, Graham Prestwich, D. Ashley Price, Charles Reynard, Beverley Riley, John Simpson, Valerie Tate, Philip Turner, Mark Wilcox, Melody Zhifang.

We acknowledge the participants for volunteering for this study and the CRN for supporting us with the sample collection and recruitment during the study, particularly Sue Dowling and Larysa Mashenko; we also acknowledge the support of the United Kingdom National Institute for Health Research Clinical Research Network and the *Co*vid-19 *N*ational *D*iagnostic *R*esearch & evaluation (CONDOR) program. In Brazil, we thank the participants and the Centro de atendimento ao colaborador, Hospital das Clínicas da Faculdade de Medicina da Universidade de São Paulo, São Paulo, Brazil.

E.A., C.E., and M.D.V. had no role in data collection and analysis. The other authors have no conflicts to declare.

This work was funded as part of FIND's work as coconvener of the diagnostics pillar of the Access to COVID-19 Tools (ACT) Accelerator, including support from Unitaid (grant no. 2019-32-FIND MDR) and the governments of the Netherlands (grant no. MINBUZA-2020.961444) and from the United Kingdom Department for International Development (grant no. 300341-102). The FALCON study was funded by the National Institute for Health Research, Asthma United Kingdom, and the British Lung Foundation. This work is partially funded by the National Institute for Health Research (NIHR) Health Protection Research Unit in Emerging and Zoonotic Infections (200907), a partnership between the United Kingdom Health Security Agency (UKHSA), The University of Liverpool, The University of Oxford, and The Liverpool School of Tropical Medicine. The views expressed are those of the author(s) and not necessarily those of the NIHR, the UKHSA, or the Department of Health and Social Care.

The study was conceived and designed by A.I.C.A., C.E., and S.F.C. Data extraction was conducted by the LSTM Diagnostic group, P.M.T., K.K., R.L.B., S.V.N., A.L.-M., A.P.M., and P.S.A. Data analysis and interpretation were conducted by R.L.B., C.R.T., M.D.V., and P.M.T. The initial manuscript was prepared by C.R.T. and P.M.T. Funding acquisition was done by A.I.C.A., R.B., the CONDOR steering group, E.A., and S.F.C. Oversight of the study was performed by A.I.C.A., the CONDOR steering group, R.B., S.F.C., and C.E. All authors edited and approved the final manuscript.

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
