## [Reviewer comments · Microbiology Spectrum]

Microbiology Spectrum

Multicentre diagnostic evaluation of OnSite COVID-19 Rapid Test (CTK Biotech) among symptomatic individuals in Brazil and The United Kingdom

Caitlin Thompson, Pablo Muñoz Torres, Konstantina Kontogianni, Rachel Byrne, LSTM Diagnostic group, Saidy Vásconez Noguera, Alessandra Luna- Muschi, Ana Paula Marchi, Pâmela Andrade, Antonio dos Santos Barboza, Marli Nishikawara, CONDOR steering group, Richard Body, Margaretha de Vos, Camille Escadafal, Emily Adams, Silvia Figueiredo Costa, and Ana Cubas-Atienzar

Corresponding Author(s): Ana Cubas-Atienzar, Liverpool School of Tropical Medicine, and Silvia Figueiredo Costa, Faculdade de Medicina da Universidade de São Paulo

Review Timeline:

Submission Date:	December 7, 2022
Editorial Decision:	January 25, 2023
Revision Received:	April 12, 2023
Accepted:	April 17, 2023

Editor: Rosemary She

Reviewer(s): Disclosure of reviewer identity is with reference to reviewer comments included in decision letter(s). The following individuals involved in review of your submission have agreed to reveal their identity: Ghulam Abbas (Reviewer #2)

Transaction Report:

DOI: <https://doi.org/10.1128/spectrum.05044-22>

January 25, 2023

Dr. Ana I. Cubas-Atienzar
Liverpool School of Tropical Medicine
Liverpool
United Kingdom

Re: Spectrum05044-22 (Multicentre diagnostic evaluation of OnSite COVID-19 Rapid Test (CTK Biotech) among symptomatic individuals in Brazil and The United Kingdom)

Dear Dr. Ana I. Cubas-Atienzar:

Thank you for submitting your manuscript to Microbiology Spectrum. The manuscript has been reviewed by two experts in the field and we have reached a decision of Modifications. When submitting the revised version of your paper, please provide (1) point-by-point responses to the issues raised by the reviewers as file type "Response to Reviewers," not in your cover letter, and (2) a PDF file that indicates the changes from the original submission (by highlighting or underlining the changes) as file type "Marked Up Manuscript - For Review Only". Please use this link to submit your revised manuscript - we strongly recommend that you submit your paper within the next 60 days or reach out to me. Detailed instructions on submitting your revised paper are below.

Link Not Available

Sincerely,

Rosemary She

Journals Department
Reviewer comments:

Reviewer #1 (Comments for the Author):

Major comments for authors

In this manuscript, Thompson et al. conducted a multicentre diagnostic evaluation of OnSite COVID-19 Rapid Test (CTK Biotech) among symptomatic individuals in Brazil and the United Kingdom. The OnSite Ag-RDT demonstrated a lower clinical sensitivity than claimed by the manufacturer. Sensitivity and specificity from the Brazil study fulfilled the performance criteria determined by the World Health Organisation but the performance obtained from the UK study failed to. There are some typos and mistakes in data analysis across the manuscript. Some suggestions can be found below:

1. Methods: Line 112-114: "The mutation panel was customized to detect the variants: Alpha (P681H), Beta (E484k + K417N),

Gamma (E484K + T20N), Delta and Kappa (L425R + P681R), Zeta (E484K) and Lambda (L452Q)." Using single mutation to differentiate different variants may be not reliable as different variants may share the same mutations, e.g., P681H shared by B.1.1.529 (Omicron) and B.1.1.7 (UK). Please provide references to justify this statement in 112-114.

2. Methods: Line 138-146: Analytical sensitivity for SARS-CoV-2 wild type (Pango, B1), Alpha (B.1.1.7), Delta (B.1.617.2), Gamma (P.1) and Omicron (B.1.1.529) were tested. It seems that Beta variant, B.1.351 (S. Africa) is missed. Please clarify.

3.1. Table 1: multiple typos/errors were found as below:

Symptoms present for Brazil: 494/495=99.8% rather than 99.6% as indicated in the Table.

Gender for UK: "52.1% (110/210)" should be "52.4% (110/210)".

Vaccinated for UK: "132, 62%": should be 62.6% or 63% (132/211) depending on how many decimal places the author prefer.

Definitely 62% indicated in the Table 1 is inaccurate.

Vaccination not disclosed (n, %) for UK : "1%" should be 0.47% (1/211)

3.2. The total numbers for calculation of different indexes were different, e.g., 496 for age whereas 495 for gender for Brazil, 211 for age whereas 210 for gender for UK. Please explain.

3.3. There are different formats of data presentation in the Table 1, e.g., "7, 1%" and "6.5%, (32/496)".

Some percentages have one decimal place (e.g., 99.6%) while others have no decimals (e.g., 4%). The format in the Table 1 should be consistent among different rows.

3.4. Please correct the corresponding parts in the main text.

4.1. Table 2: invalid rate: 0.6% (3/496): what does "496" come from? There are a lot of 495 samples in Brazil, right? Please clarify.

4.2. Table 2: the unexpected lower sensitivity of the OnSite test may be due to a limited number of PCR-confirmed positive samples enrolled in this study, e.g., 31 PCR positive samples in Brazil and 77 PCR positive samples in UK. The authors should test more PCR-confirmed positive samples by the OnSite test to further examine the clinical sensitivity.

5.1. Table 3: There are different formats of data presentation in the Table 3, e.g., "1, 3%" and "90.5%, 42".

Some percentages have one decimal place (e.g., 95.0%) while others have no decimals (e.g., 1%). The format in the Table 3 should be consistent among different rows.

5.2. Ct > 25 (n, %) : 7, 22% should be 22.6% or 23% (7/31) depending on how many decimal places the author prefer. Definitely 22% indicated in the Table 3 is inaccurate.

6. Statistical analysis: please clarify the statistical analysis were two-tailed or one-tailed.

Minor comments for authors

7. Line 46 "clinical sensitivity than claimed by the manufacturer..." Please change "... to "."

8. Line 58-60 "However, since April 2022 the UK government has ceased free Ag-RDT testing, now requiring the responsibility of the acquisition, and performance of tests to be placed on the individual." What does it mean by "now requiring the responsibility of the acquisition, and performance of tests to be placed on the individual"? Please clarify.

9. Line 123-125: "Swabs were taken systematically, NP swab samples in UTM (Copan Diagnostics Inc, Italy) were collected for the reference RT-qPCR test, this was followed by an NP swab to perform the Ag-RDTs." This sentence has grammatical issue so please re-write it. In addition, what does it mean by "swabs were taken systematically"? Please clarify.

10. Line 126-127: "All samples were transported in cooler boxes to the Liverpool School of Tropical Medicine (LSTM) and processed upon..." What is the shipment conditions and temperature in the cooler boxes?

Reviewer #2 (Comments for the Author):

This is an important research and met analysis of Multicentre diagnostic evaluation of OnSite COVID-19 Rapid Test (CTK Biotech) among symptomatic individuals in Brazil and The United Kingdom, it pulls the data together in a way that is likely to be highly impactful and provides a state of the art overview of the current state of knowledge on analytical evaluation of the Ag-RDT. It carries key information about the Ag-RDT assay using direct culture supernatant of SARS-CoV-2 strains from Wild-Type (WT), Alpha, Delta, Gamma, and Omicron lineages, and is likely to be highly cited in the future. However, the manuscript needs to be improved for certain necessary changes and spelling/grammar mistakes throughout and I support its publication after some minor changes. I would like to advise the author to strictly follow the journal's guidelines such as reference style etc.

1-Give full names of all the abbreviations used first, then you can use abbreviations.

The English language of the article also needs to be improved

Italicize all the "et al., or et al." and scientific words (line 105).

The discussion part needs a more detailed analysis of the results.

Staff Comments:

Preparing Revision Guidelines

Please return the manuscript within 60 days; if you cannot complete the modification within this time period, please contact me. If you do not wish to modify the manuscript and prefer to submit it to another journal, please notify me of your decision immediately so that the manuscript may be formally withdrawn from consideration by Microbiology Spectrum.

Multicentre diagnostic evaluation of OnSite COVID-19 Rapid Test (CTK Biotech) among symptomatic individuals in Brazil and The United Kingdom

Caitlin R Thompson¹, Pablo Muñoz Torres ², Konstantina Kontogianni¹, Rachel L Byrne¹, LSTM Diagnostic group, Saidy Vásquez Noguera^{2,3}, Alessandra Luna-Muschi^{2,3}, Ana Paula Marchi^{2,3}, Pâmela S Andrade⁴, Antonio dos Santos Barboza⁵, Marli Nishikawara⁵, CONDOR steering group, Richard Body⁸, Margaretha de Vos⁷, Camille Escadafal⁶, Emily Adams^{1,9}, Silvia Figueiredo Costa^{2,3*}, Ana I Cubas Atienzar^{1*}

¹ Liverpool School of Tropical Medicine, Centre for Drugs and Diagnostics, Liverpool, UK

² LIM-49, Instituto de Medicina Tropical, Faculdade de Medicina da Universidade de São Paulo, São Paulo, Brazil

³ Departamento de Moléstias Infecciosas e Parasitárias, Faculdade de Medicina da Universidade de São Paulo, São Paulo, Brazil

⁴ Department of Epidemiology, School of Public Health of University of São Paulo, São Paulo, Brazil.

⁵ Centro de atendimento ao colaborador, Hospital das Clínicas da Faculdade de Medicina da Universidade de São Paulo, São Paulo, Brazil.

⁶ Divisão de Laboratório Central, Hospital das Clínicas, Faculdade de Medicina da Universidade de São Paulo, São Paulo, Brazil.

⁷ FIND, Geneva, Switzerland

⁸ Manchester University NHS Foundation Trust, UK

⁹ Global access diagnostics, Bedfordshire, UK

* Joint last senior authors

Corresponding authors: Ana I Cubas Atienzar Ana.CubasAtienzar@lstmed.ac.uk (UK) and Silvia Figueiredo Costa silviacosta@usp.br (Brazil)

ABSTRACT (305 words)

The COVID-19 pandemic has given rise to numerous commercially available antigen rapid diagnostic tests (Ag-RDTs). To generate and share accurate and independent data with the

global community, multi-site prospective diagnostic evaluations of Ag-RDTs are required. This report describes the clinical evaluation of the OnSite COVID-19 Rapid Test (CTK Biotech, California, USA) in Brazil and The United Kingdom.

A total of 496 paired nasopharyngeal (NP) swabs were collected from symptomatic healthcare workers at Hospital das Clínicas in São Paulo, and 211 NP swabs were collected from symptomatic participants at a COVID-19 drive-through testing site in Liverpool, England. These swabs were analyzed by Ag-RDT and results were compared to RT-qPCR.

The clinical sensitivity of the OnSite COVID-19 Rapid test in Brazil was 90.3% [95% CI 75.1 – 96.7%] and in the United Kingdom was 75.3% [95% CI 64.6 – 83.6%]. The clinical specificity in Brazil was 99.4% [95% CI 98.1 – 99.8%] and in the United Kingdom was 95.5% [95% CI 90.6 – 97.9%]. Concurrently analytical evaluation of the Ag-RDT was assessed using direct culture supernatant of SARS-CoV-2 strains from Wild-Type (WT), Alpha, Delta, Gamma, and Omicron lineages.

This study provides a comparative performance of an Ag-RDT across two different settings, geographical areas, and populations. Overall, the OnSite Ag-RDT demonstrated a lower clinical sensitivity than claimed by the manufacturer... Sensitivity and specificity from the Brazil study fulfilled the performance criteria determined by the World Health Organisation but the performance obtained from the UK study failed to. Further evaluation of the use of Ag-RDTs should include harmonized protocols between laboratories to facilitate comparison between settings.

Introduction

To meet the immense diagnostic demand of the COVID-19 pandemic, the use of rapid diagnostic tests for the detection of SARS-CoV-2 antigens (Ag-RDTs) has become a priority. To date, there are currently 321 SARS-CoV-2 Ag-RDTs on the market or in development according to the foundation for new innovative diagnostics (FIND) (date accessed March 2022)[1]. However, clinical evaluation of these Ag-RDTs has been relatively limited and performance results differ greatly between studies[2, 3]. In the UK, the use of Ag-RDTs has been integral to reducing the spread of COVID-19 [4]. However, since April 2022 the UK government has ceased free Ag-RDT testing, now requiring the responsibility of the acquisition, and performance of tests to be placed on the individual.

In Brazil, the national SARS-CoV-2 testing approach has been insufficient in its use of this diagnostic tool in the efforts to contain this pandemic [5]. Many initiatives such as recruiting capacity in university research laboratories and biotechnological enterprises, investments in new laboratory infrastructure, and fast-track regulatory measures were launched to scale up SARS-CoV-2 RT-qPCR testing in Brazil. However, the expansion of the quantitative reverse transcriptase polymerase chain reaction (RT-qPCR) capacity has not been sufficient to control the progress of the pandemic within this country [5].

Despite the commercialization of several vaccines for SARS-CoV-2, the COVID-19 pandemic is still ongoing due to vaccine inequity [6], uneven vaccine uptake between populations [7], and the emergence of new SARS-CoV-2 highly transmissible variants [8].

The gold standard for diagnosis of COVID-19 remains the detection of SARS-CoV-2 ribonucleic acid (RNA). However, RT-qPCR requires skilled laboratory scientists, installed capacity, and expensive consumables and reagents which can be challenging to implement in low and middle-income countries (LMIC), where the burden of COVID-19 is disproportionately felt. Additionally, the turnaround of results of RT-qPCR can take up to one week [9].

To continue to meet the challenges of testing capacity, prospective diagnostic evaluation studies across multiple, independent sites are required to determine the accuracy of COVID-19 Ag-RDTs available for purchase to the public.

In this study, OnSite COVID-19 Rapid Test (CTK Biotech) was evaluated against the SARS-CoV-2 diagnostic gold standard RT-qPCR. Testing was undertaken in Brazil and the UK across different settings: on healthcare workers (HCWS) at Hospital das Clínicas, a tertiary-care hospital affiliated with the University of São Paulo (Brazil), and at a National Health Service COVID-19 drive-through community testing center in Liverpool, UK.

Methods

Clinical evaluation

This was a prospective evaluation of consecutive participants enrolled in two different settings:

Brazil

Healthcare workers (HCW) with suspected COVID-19 symptoms (fever, cough, shortness of breath, tight chest, runny nose, sore throat, anosmia, ageusia, headache, and diarrhea) were enrolled at the HCW service of Hospital das Clínicas in São Paulo from July to October 2021. Ethical approval was obtained from the Hospital's Ethics Committee with the CAAE number 35246720.0.0000.0068. Informed consent was obtained from all study participants for respiratory samples and clinical data collection.

Participants were clinically evaluated and RT-qPCR for SARS-CoV-2 was performed from combined nasopharyngeal and oropharyngeal swabs (Goodwood Medical Care LTD/(DG) China) following the national standard of care. Following the RT-qPCR swabs, nasopharyngeal (NP) swabs were collected for Ag-RDT testing. The OnSite Ag-RDT was performed at the point-of-care by HCW following the manufacturer's instructions for use (IFU).

For SARS-CoV-2 RT-qPCR, RNA was extracted from saline solution 0.9% with an automated method using magnetic beads (Sample Preparation System RNA, Abbott, Illinois, USA). SARS-CoV-2 RT-qPCR was performed using an adapted protocol described by Corman, Victor M et al, 2020 [10] to detect the E gene as the first-line screening tool, followed by confirmatory testing with an assay detecting the N gene (Abbott, USA) and the commercial SARS-CoV-2 N1+N2 RT-qPCR kit to detect N1 and N2 genes (Qiagen, USA). SARS-CoV-2 RT-qPCR result was considered positive with an amplification cycle threshold (Ct) ≤ 32 and (Ct) ≤ 33 , respectively.

Positive samples underwent genotyping for variant identification using TaqPath™ 1-Step RT-qPCR Master SARS-CoV-2 Mutation Panel Assay (Thermo Fisher Scientific, Waltham, USA). Data were analyzed by QuantStudio™ v2.5.1 design and analysis software in the genotyping module following IFU. The mutation panel was customized to detect the variants: Alpha (P681H), Beta (E484k + K417N), Gamma (E484K + T20N), Delta and Kappa (L425R + P681R), Zeta (E484K) and Lambda (L452Q).

United Kingdom (UK)

In the UK, adults presenting with symptoms of COVID-19 (fever, cough, shortness of breath, tight chest, runny nose, sore throat, anosmia, ageusia, headache, diarrhea, and tiredness) at a national community testing facility, the Liverpool John Lennon Airport drive-through

COVID-19 test center, were asked to participate in the study. Participants were recruited between July and August of 2021 under the Facilitating Accelerated COVID-19 Diagnostics (FALCON) study. Ethical approval was obtained from the National Research Ethics Service and the Health Research Authority (IRAS ID:28422, clinical trial ID: NCT04408170).

Swabs were taken systematically, NP swab samples in UTM (Copan Diagnostics Inc, Italy) were collected for the reference RT-qPCR test, this was followed by an NP swab to perform the Ag-RDTs. Due to biosafety restrictions at the drive-through center, Ag-RDT testing was not done immediately after sample collection as per the IFU. All samples were transported in cooler boxes to the Liverpool School of Tropical Medicine (LSTM) and processed upon arrival by trained laboratory researchers following the IFU. Processing happened maximum within 3 hours of collection. Ag-RDTs were performed and the UTM NP swab samples were aliquoted and stored at -80°C until RNA extraction. RNA was extracted using the QIAamp® 96 Virus QIAcube® HT kit (Qiagen, Germany) on the QIAcube® (Qiagen, Germany) and screened using TaqPath COVID-19 (ThermoFisher, UK) on the QuantStudio 5™ thermocycler (ThermoFisher, UK). SARS-CoV-2 RT-qPCR result was considered positive if any two of the three targets (N, ORFab, and S) were amplified with cycle threshold (Ct) ≤ 40 .

Analytical Sensitivity (UK only)

Viral culture methods to propagate SARS-CoV-2 isolates and to calculate plaque forming units per milliliter (PFU/mL) followed that previously described [11]. Briefly, isolates of SARS-CoV-2 from the wild type (Pango, B.1) (REMRQ0001/Human/2020/Liverpool, GISAID ID EPI_ISL_464183), Alpha (B.1.1.7) (SARS-CoV-2/human/GBR/FASTER_272/2021, GenBank ID MW980115), Delta (B.1.617.2) (SARS-CoV-2/human/GBR/Liv_273/2021, GenBank ID OK392641), Gamma (P.1) (hCoV-19/Japan/TY7-503/2021, GISAID ID EPI_ISL_792683) and Omicron (B.1.1.529) (hCoV-19/USA/MD-HP20874/2021, GISAID ID EPI_ISL_7160424) lineages were used to evaluate the limit of detection (LOD) of the OnSite Ag-RDT. For the determination of the LOD, a fresh aliquot was serially diluted from 1.0×10^5 plaque-forming units (pfu)/mL to 1.0×10^2 pfu/mL. Each dilution was tested in triplicate. Two-fold dilutions were made below the ten-fold LOD dilution to confirm the lowest LOD (LLOD).

Viral RNA was extracted from each dilution using QIAamp Viral RNA mini kit (Qiagen, Germany) according to the manufacturer's instructions, and quantified using Genesig RT-

qPCR (Primer Design, UK). Genome copy number/mL (gcn/mL) was calculated as previously described [12].

Statistical Analysis

The sensitivity and specificity, with 95% confidence intervals (CIs) were calculated based on the results of the reference method by RT-qPCR assay. Statistical analyses were performed using R scripts, Epi Info, and GraphPad Prism 9.1.0 (GraphPad Software, Inc, California). The 95% confidence interval (CI) for the sensitivity and specificity was calculated using Wilson's method. Fisher's exact and chi-squared tests were used to determine non-random associations between categorical variables. Statistical significance was set at $P < 0.05$.

Results

Clinical Evaluation

The demographics of both the Brazilian and UK study cohorts are shown in Table 1. In Brazil, the median days from onset of symptoms was 3 [Q1-Q3, 2-4], with a vaccination rate of 97.0% (including partial and fully vaccinated participants). In the UK the median days from symptom onset was 2 [Q1-Q3, 1-3] and the vaccination rate was 84.4% (including partial and fully vaccinated participants). Significantly higher SARS-CoV-2 RT-qPCR positivity was detected in the UK (36.5%, CI 0.29-0.43) than in Brazil (6.5%, CI 0.05-0.09) ($P < 0.05$).

The clinical sensitivity of the Onsite Ag-RDT across evaluation sites was heterogeneous, with a clinical sensitivity of 90.3% [95% CI 75.1-96.7%] in Brazil and 75.3% [95% CI 64.6-83.6%] in the UK (Table 2). The difference in sensitivities between sites was not statistically significant ($P = 0.128$). The clinical specificity of the Onsite Ag-RDT was 99.4% [95% CI 98.1-99.8%] in Brazil and 95.5% [95% CI 90.6-97.9%] in the UK.

In Brazil, of the 496 participants included, 32 were SARS-CoV-2 RT-qPCR positive (6.5%) (see Table 2). Twenty-eight of the RT-qPCR positive samples (90.3%) were Ag-RDT positive, while 3 (9.7%) were Ag-RDT negative and one was invalid (3.1%). Invalid results were removed for further analysis. Of the 464 RT-qPCR negative samples, 3 were Ag-RDT positive (0.6%). The sensitivity and specificity of the OnSite Ag-RDT test on RT-qPCR were 90.3% [95% CI 75.1%-96.7%] and 99.4% [95% CI 98.1%-99.8%], respectively (See Table 2). Sensitivity ≤ 7 days symptom onset was 96.2% [95% CI 81.1-99.3%]. Sensitivity according to Ct value was 95%

[95% CI 75.1-99.8%] for Ct \leq 25 and 90.3% [95% CI 75.1%-96.7%] for Ct \leq 33 (See Table 3). No statistical significance was found in sensitivity between different Ct value groups.

In the UK, of the 211 participants recruited, 77 (36.5%) were SARS-CoV-2 RT-qPCR positive (see Table 2). Fifty-Eight (75.3%) of the 77 RT-qPCR positive samples were also Ag-RDT positive, while 19 (24.7%) were Ag-RDT negative. Of the 134 RT-qPCR negative samples, 128 (95.5%) were also Ag-RDT negative and 6 (4.5%) were Ag-RDT positive. For the UK evaluation, the sensitivity and specificity were 75.3% [95% CI 64.6-83.6%] and 95.5% [95% CI 90.6-97.9%], respectively. Sensitivity \leq 7 days symptom onset was 76.7% [95% CI 65.8-84.9%]. Ct values of \leq 20, \leq 25, \leq 33, and \leq 40 had a sensitivity of 90.5% [95% CI 77.4%-97.3%], 80.3% [95% CI, 69.2%-88.1%], 76.3% [95% CI 65.5-84,5%] and 75.3% [95% CI 64.6-83.6%] respectively. Sensitivity was statistically higher among samples with Ct values \leq 20 compared with samples with Ct values \leq 33 ($P = 0.029$) and \leq 40 ($P = 0.044$).

Subgroup analyses of the Brazilian and UK evaluation cohorts (Table 4) were performed to determine any associated differences in sensitivity compared to vaccination status and days from symptom onset. In the Brazilian cohort, the sensitivity of the OnSite Ag-RDT was significantly lower on samples from patients with symptoms onset >7 days compared to samples with 0-3 symptoms onset ($P = 0.02924$) and samples with 0-7 days of onset ($P = 0.03115$) but no differences in sensitivity were found between groups of different vaccination status. In the UK, no difference in sensitivity was observed between groups of different symptoms onset and vaccination status (all P values >0.05). In Brazil, 52% of the positive samples were classified as Delta and 39% Gamma. In the UK, the variant determination was not performed but at the time of enrollment, 100% of genome submissions corresponded to the Delta variant [13].

Table 1**Demographics of Ag-RDT clinical evaluation cohorts for Brazil and United Kingdom**

Country	Brazil	United Kingdom
Age [mean (min-max), N]	38.1 (16-69), 496	40.8 (20-86), 211
Gender [%F, (n/N)]	71.5 % (354/495)	52.1% (110/210)
Symptoms present [%Yes, n/N]	99.6% (494/495)	100% (211/211)
Days from symptom onset [median (Q1-Q3); N]	3 (2-4), 494	2 (1-3), 211
Days 0-3 (n, %)	294, 60%	169, 80%
Days 4-7 (n, %)	186, 38%	36, 17%
Days 8+ (n, %)	14, 3%	6, 3%
Vaccinated (n, %)	460, 93%	132, 62%
Partially Vaccinated (n, %)	19, 4%	47, 22%
Not vaccinated (n, %)	10, 2%	32, 15%
Vaccination not disclosed (n, %)	7, 1%	1, 1%%
SARS-CoV-2 Positivity [%, (n/N)]	6.5%, (32/496)	36.5%, (77/211)

Table 2

Results and clinical sensitivity and specificity of the OnSite COVID-19 Ag Device based on COVID-19 RT-qPCR result in Brazil and the United Kingdom

Results of OnSiteCOVID-19 Ag Device	Brazil			United Kingdom		
	Confirmed by RT-qPCR					
	Positive	Negative	Total	Positive	Negative	Total
Positive	28	3	31	58	6	64
Negative	3	461	464	19	128	147
Total	31	464	495	77	134	211
Clinical Sensitivity (95% CI), N	90.3% (75.1-96.7%), 31			75.3% (64.6-83.6%), 77		
Clinical Specificity (95% CI), N	99.4% (98.1-99.8%), 464			95.5% (90.6-97.9%), 134		
Invalid Rate (% , n/N)	0.6% (3/496)			0% (211/211)		

RT-qPCR = Real-time quantitative polymerase chain reaction, Ct = cycle threshold, CI = confidence interval

Table 3**COVID-19 RT-qPCR result in Brazil and the United Kingdom**

	Brazil	United Kingdom
PCR Ct [median (Q1-Q3); N]	19.6 (17.52-23), 31	19.5 (17.3-22.8), 77
Ct > 33 (n, %)	0, 0%	1, 1%
Ct > 30 (n, %)	1, 3%	5, 6%
Ct > 25 (n, %)	7, 22%	11, 14%
Sensitivity Ct ≤20, N	100%, (76.8-100%),14	90.5% (77.4-97.3%), 42
Sensitivity Ct ≤25, N	95.0% (75.1-99.8%), 20	80.3% (69.2-88.1%), 66
Sensitivity Ct ≤33, N	90.3% (75.1-96.7%), 31	76.3% (65.5-84.5%), 76
Sensitivity Ct ≤40, N	NA*	75.3% (64.6-83.6%), 77

* Maximum RT-qPCR cut off was ≤33 in Brazil

Table 4

Ag-RDT results by the onset of symptoms, and vaccinated individuals in Brazil and the UK

	Brazil				United Kingdom			
	Ag-RDT Positive (n, %)	Ag-RDT Negative (n, %)	Sensitivity	95% CI	Ag-RDT Positive (n, %)	Ag-RDT Negative (n, %)	Sensitivity ^a	95% CI
Days from symptom onset								
Days 0-3	16, 5.4%	278, 94.6%	100.0%	76.9-100.0%	52, 30.6%	117, 69.4%	79.7%	67.2-89.0%
Days 4-7	12, 6.4%	173, 93.6%	91.7%	61.5-99.8%	10, 27.8%	26, 72.2%	64.3%	35.2-87.3%
Days 8+	3, 21.4%	11, 78.6%	60.0%	14.7-94.7%	2, 33.3%	4, 66.7%	66.7%	9.4-99.2%
Vaccination received								
Vaccinated ^b	31, 6.5%	447, 93.5%	93.3%	77.4-99.2%	52, 29.3%	126, 70.7%	78.3%	65.8-87.9%
Not vaccinated	1, 10.0%	9, 90.0%	0.00%	N/A	11, 34.4%	21, 65.6%	62.5%	35.4-84.8%
Not disclosed	0, 0.0%	7, 100.0%	N/A	N/A	1, 100.0%	0, 0.0%	100%	2.5-100.0%

^a As compared to RT-qPCR

^bVaccinated defined as 1 or more doses

RT-qPCR = Real-time quantitative polymerase chain reaction, CI = confidence interval

Analytical sensitivity

The LOD of the OnSite Ag-RDT was 1.0×10^3 pfu/mL, 1.0×10^3 pfu/mL, 1.0×10^2 pfu/mL, 5.0×10^3 pfu/mL and 1.0×10^3 pfu/mL when tested on the WT, Alpha, Delta, Gamma and Omicron lineages, respectively. This gave a viral copy equivalent of approximately 2.1×10^5 copies/mL, 2.1×10^4 copies/mL, 1.6×10^4 copies/mL, 3.5×10^6 copies/mL, and 8.7×10^4 copies/mL for the Ag-RDT for the WT, Alpha, Delta, Gamma and Omicron lineages.

Discussion

The study aimed to evaluate the diagnostic performance of the OnSite COVID-19 Ag Rapid Test (CTK Biotech) in two different settings. Evaluating rapid diagnostic tests in diverse populations is vital to improving diagnostic responses as it indicates the diagnostic accuracy in real-world scenarios. In the case of rapid diagnostic testing within this pandemic, lateral flow tests which meet the minimum requirements for sensitivity and specificity can play a key role in increasing testing capacity, allowing timely clinical management of those infected and protecting healthcare systems [14]. This is particularly valuable in settings where access to the gold-standard RT-qPCR is often not available. Ag-RDTs are low-cost, easy to use, and do not require specialized skills or equipment which is essential to promote universal access.

The sensitivity and specificity of the OnSite Ag-RDT in a hospital setting in Brazil fulfilled the performance criteria determined by the World Health Organisation (WHO). However, sensitivity obtained in a community setting at a drive-through testing site in the UK missed the minimum recommendations [15] for both sensitivity and specificity. In guidance published by the WHO, minimum performance requirements for an Ag-RDT include a sensitivity of $>80\%$ and specificity of $>97\%$ [15]. Analytical evaluation of OnSite Ag-RDT detected Wild Type, Alpha, Delta, and Omicron, meeting the recommendations in the WHO Target Product Profile for SARS-CoV-2 Ag-RDT of an acceptable analytical LOD of 1.0×10^6 RNA copies/mL [16] with the Gamma variant, slightly outside this threshold. In the Brazilian cohort, the Gamma variant was responsible for 39% of infections and the Delta variant was responsible for 52%. This is an interesting finding as it does not reflect the wider variant circulation in Brazil during this period as the Gamma variant was responsible for over 93% of infections in July 2021 and 70% of infections in August 2021 followed by Delta at 5% rising to 29% respectively [17]. In the UK, positive RT-qPCR results were not sequenced but it is

assumed that all infections were Delta (B.1.617.2) due to the >99% circulation of this variant in the UK during the time of collection [18].

This study has several strengths, it is a multicentre and multinational evaluation across two different settings with differing testing capacities, the prevalence of SARS-CoV-2, and population characteristics. In Brazil, samples were taken from a very exclusive population, healthcare workers in a healthcare setting with a high vaccination uptake compared to the rest of the population [19]. In the UK, data was collected from a diverse population, any person over the age of 18 presenting with COVID-19 symptoms at a government-run, drive-through COVID-19 testing facility. It is important to evaluate Ag-RDTs in a heterogeneous population and setting to obtain meaningful diagnostic accuracy data.

The main limitation of the study is that the drive-through testing setting in the UK did not allow for Ag-RDT testing to be performed at the point-of-care just after sample collection as recommended by the IFU. Guidance in the UK restricted testing of suspected COVID-19-positive individuals to high containment laboratories. Currently, there are limited studies on the stability of Ag-RDT's. A systematic review of Ag-RDTs did not find a significant difference between Ninety-six data sets that involved fresh specimens for antigen testing, and 23 data sets that included freeze/thawed specimens for antigen testing [20] Although it is not stated whether the swabs were frozen dried or using transport buffer. However, one review of Ag-RDT performance in sub-Saharan Africa suggested that a delay in performing the test (CORIS COVID-19 Ag Respi-strip) may impact its stability if stored at 4°C rather than frozen at -20°C immediately [21]. Conversely, studies have shown that SARS-CoV-2 RNA remains stable for up to 9 days in dry swabs at the ambient temperature of 20°C [22] and proteins are shown to be more stable than RNA [23]. Therefore, further investigation must take place to determine whether the time from sample collection to Ag-RDT testing has a significant impact on the sensitivity.

Two other limitations of this study are that the RT-qPCR methodologies varied between both cohorts and the differences in SARS-CoV-2 prevalence. These factors have been attributed to a major cause of index case diagnostic accuracy [24]. For future evaluations, quantification of the viral copy numbers rather than Ct values is recommended to mitigate differences in RT-qPCR assay performances. This Ct variability has been estimated to be >

1000-fold in viral copy numbers/mL [24], as the RT-qPCR used in the UK has a LOD 10-fold more sensitive (10 genome copies/mL) than the RT-qPCR used in Brazil (100 genome copies/mL) [25]. The higher sensitivity of the RT-qPCR assay used in the UK, together with the higher cut-off used (Ct 40 versus Ct 32-33 in Brazil) could have contributed to higher numbers of false negatives in the index test compared to the Brazilian cohort. Additionally, there is a significant difference in sample size and in confirmed RT-qPCR positives (SARS-CoV-2 prevalence) between the two cohorts, with a low number of positive samples found in the Brazilian evaluation (6.3%) compared to the UK (36.5%). It has been reported that differences in prevalence can affect the sensitivity and specificity of index tests [26, 27].

In conclusion, the data indicate that OnSite Ag-RDT had lower performance quality than published by the manufacturers for the detection of SARS-CoV-2 in clinical samples and varied greatly between the two settings in this study. Further evaluation of the use of Ag-RDTs should strictly follow the IFUs of the test and include harmonized protocols between laboratories to facilitate comparison between settings. In particular, the use of viral copy numbers rather than Ct values has been suggested to minimize the variability between laboratories.

Acknowledgments

LSTM diagnostics group: Ms Caitlin Greenland Bews, Ms Kate Buist, Ms Karina Clerkin, Dr Thomas Edwards, Dr Lorna Finch, Dr Helen R Savage, Ms Jahanara Wardale, Ms Rachel Watkins, Mr Chris Williams and Mr Dominic Wooding.

Condor steering group: Dr A. Joy Allen, Dr Julian Braybrook, Professor Peter Buckle, Ms Eloise Cook, Professor Paul Dark, Dr Kerrie Davis, Dr Gail Hayward, Professor Adam Gordon, Ms Anna Halstead, Dr Charlotte Harden, Dr Colette Inkson, Ms Naoko Jones, Dr William Jones, Professor Dan Lasserson, Dr Joseph Lee, Dr Clare Lendrem, Dr Andrew Lewington, Ms Mary Logan, Dr Massimo Micocci, Dr Brian Nicholson, Professor Rafael Perera-Salazar, Mr Graham Prestwich, Dr D. Ashley Price, Dr Charles Reynard, Dr Beverley Riley, Professor John Simpson, Dr Valerie Tate, Dr Philip Turner, Professor Mark Wilcox, Dr Melody Zhifang.

We would like to acknowledge the participants for volunteering for this study and the CRN for supporting us with the sample collection and recruitment during the study; particularly

Sue Dowling and Larysa Mashenko; we also acknowledge the support of the UK National Institute for Health Research Clinical Research Network and the COVID-19 National Diagnostic Research & evaluation (CONDOR) program. In Brazil, we would like to thank the participants and the Centro de atendimento ao colaborador, Hospital das Clínicas da Faculdade de Medicina da Universidade de São Paulo, São Paulo, Brazil.

Conflicts of Interest: EA, CE, and MDV had no role in data collection and analysis. The other authors have no conflicts to declare.

Research Funding: This work was funded as part of FIND's work as co-convenor of the diagnostics pillar of the Access to COVID-19 Tools (ACT) Accelerator, including support from Unitaid [grant number: 2019-32-FIND MDR], the governments of the Netherlands [grant number: MINBUZA-2020.961444] and from UK Department for International Development [grant number 300341-102]. The FALCON study was funded by the National Institute for Health Research, Asthma UK and the British Lung Foundation. This work is partially funded by the National Institute for Health Research (NIHR) Health Protection Research Unit in Emerging and Zoonotic Infections (200907), a partnership between the UK Health Security Agency, The University of Liverpool, The University of Oxford, and The Liverpool School of Tropical Medicine. The views expressed are those of the author(s) and not necessarily those of the NIHR, the UKHSA, or the Department of Health and Social Care.

Contributor Role	Role Definition	Authors
Conceptualization	Ideas; formulation or evolution of overarching research goals and aims.	RB, EA, CSG, AICA, CE, PMT, AJLM, SFC
Data Curation	Management activities to annotate (produce metadata), scrub data, and maintain research data (including software code, where it is necessary for interpreting the data itself) for initial use and later reuse.	MdV, SGC, RB, AICA, SVN, ASB, MN, APM, ECS, CLS
Formal Analysis	Application of statistical, mathematical, computational, or other formal techniques to analyze or synthesize study data.	CRT, AICA, PMT, AJLM, SFC
Funding Acquisition	Acquisition of the financial support for the project leading to this publication.	RB, AICA, GH
Investigation	Conducting a research and investigation process, specifically performing the experiments, or data/evidence collection.	CRT, PMT, AJLM, DB, KK, PSA, ASB
Methodology	Development or design of methodology; creation of models.	CRT, SFC
Project Administration	Management and coordination responsibility for the research activity planning and execution.	AICA, EA, CSG, SFC
Resources	Provision of study materials, reagents, materials, patients, laboratory samples, animals, instrumentation, computing resources, or other analysis tools.	EC, AICA, RB
Software	Programming, software development; designing computer programs; implementation of the computer code and supporting algorithms; testing of existing code components.	MdV, CEG
Supervision	Oversight and leadership responsibility for the research activity planning and execution, including mentorship external to the core team.	EA, AICA
Validation	Verification, whether as a part of the activity or separate, of the overall replication/reproducibility of results/experiments and other research outputs.	AICA, PMT, AJL, SFC, RB, EA, CE
Visualization	Preparation, creation, and/or presentation of the published work, specifically visualization/data presentation.	CRT, PMT, SFC, AICA, RLB
Writing – Original Draft	Creation and/or presentation of the published work, specifically writing the initial draft (including	CRT, PMT, AJLM

Preparation	substantive translation).	
Writing – Review & Editing	Preparation, creation, and/or presentation of the published work by those from the original research group, specifically critical review, commentary, or revision – including pre- or post-publication stages.	AICA, SFC

References

1. FIND. *Test Directory*,. 2022; Available from: <https://www.finddx.org/test-directory/?test-format=lateral-flow-assay-strip-or-cassette&assay-target=antigen>.
2. Dinnes, J., et al., *Rapid, point-of-care antigen and molecular-based tests for diagnosis of SARS-CoV-2 infection*. Cochrane Database of Systematic Reviews, 2021(3).
3. Krüger, L.J., et al., *Accuracy and ease-of-use of seven point-of-care SARS-CoV-2 antigen-detecting tests: A multi-centre clinical evaluation*. eBioMedicine, 2022. **75**: p. 103774.
4. World Health Organization, *Recommendations for national SARS-CoV-2 testing strategies and diagnostic capacities*. 2021.
5. Kameda, K., et al., *Testing COVID-19 in Brazil: fragmented efforts and challenges to expand diagnostic capacity at the Brazilian Unified National Health System*. Cad Saude Publica, 2021. **37**(3): p. e00277420.
6. Pilkington, V., S.M. Keestra, and A. Hill, *Global COVID-19 Vaccine Inequity: Failures in the First Year of Distribution and Potential Solutions for the Future*. Frontiers in Public Health, 2022. **10**.
7. Mathieu, E., et al., *A global database of COVID-19 vaccinations*. Nature Human Behaviour, 2021. **5**(7): p. 947-953.
8. World Health Organization. *Tracking SARS-CoV-2 variants*. 2022 [cited 2022 28/01/2022]; Available from: <https://www.who.int/en/activities/tracking-SARS-CoV-2-variants/>.
9. Berti, L. *Patients in Sao Paulo waiting a week for Covid-19 test results*. 2022; Available from: <https://brazilian.report/liveblog/2022/01/21/patients-sao-paulo-test-results/>.
10. Corman, V.M., et al., *Detection of 2019 novel coronavirus (2019-nCoV) by real-time RT-PCR*. Eurosurveillance, 2020. **25**(3): p. 2000045.
11. Edwards, T., et al., *SARS-CoV-2 Transmission Risk from sports Equipment (STRIKE)*. medRxiv, 2021: p. 2021.02.04.21251127.
12. Cubas-Atienzar, A.I., et al., *Limit of detection in different matrices of 19 commercially available rapid antigen tests for the detection of SARS-CoV-2*. Scientific Reports, 2021. **11**(1): p. 18313.
13. Next Strain. *Genomic epidemiology of novel coronavirus - Global subsampling*. 2022; Available from: https://nextstrain.org/ncov/gisaid/global?f_country=United%20Kingdom.
14. Peeling, R.W., et al., *Diagnostics for COVID-19: moving from pandemic response to control*. The Lancet, 2022. **399**(10326): p. 757-768.
15. World Health Organization, *Coronavirus disease 2019 (COVID-19): situation report*,. 2020, World Health Organization.
16. World Health Organization, *COVID-19 Target product profiles for priority diagnostics to support response to the COVID-19 pandemic v.1.0*. 2020.
17. Our World Data. *SARS-CoV-2 variants in analyzed sequences*. 2022; Available from: <https://ourworldindata.org/grapher/covid-variants-area?country=~BRA>.
18. Our World Data. *SARS-CoV-2 variants in analyzed sequences, United Kingdom*. 2022 [cited 2022; Available from: <https://ourworldindata.org/grapher/covid-variants-area?country=~GBR>.

19. Coronavirus Brazil. *Painel Coronavírus Atualizado*. 2022; Available from: <https://covid.saude.gov.br/>.
20. Parvu, V., et al., *Factors that Influence the Reported Sensitivity of Rapid Antigen Testing for SARS-CoV-2*. *Frontiers in Microbiology*, 2021. **12**.
21. Jacobs, J., et al., *Implementing COVID-19 (SARS-CoV-2) Rapid Diagnostic Tests in Sub-Saharan Africa: A Review*. *Front Med (Lausanne)*, 2020. **7**: p. 557797.
22. Gokulan, C.G., et al., *Temporal stability and detection sensitivity of the dry swab-based diagnosis of SARS-CoV-2*. *Journal of Biosciences*, 2021. **46**(4): p. 95.
23. *How fast do RNAs and Proteins degrade?* Cell biology by the numbers; Available from: <http://book.bionumbers.org/how-fast-do-rnas-and-proteins-degrade/>.
24. Evans, D., et al., *The Dangers of Using C_q to Quantify Nucleic Acid in Biological Samples: A Lesson From COVID-19*. *Clin Chem*, 2021. **68**(1): p. 153-162.
25. Sohni, Y., *Variation in LOD Across SARS-CoV-2 Assay Systems: Need for Standardization*. *Laboratory Medicine*, 2020. **52**(2): p. 107-115.
26. Leeflang, M.M.G., P.M.M. Bossuyt, and L. Irwig, *Diagnostic test accuracy may vary with prevalence: implications for evidence-based diagnosis*. *Journal of Clinical Epidemiology*, 2009. **62**(1): p. 5-12.
27. Leeflang, M.M., et al., *Variation of a test's sensitivity and specificity with disease prevalence*. *Cmaj*, 2013. **185**(11): p. E537-44.

Caitlin Thompson
Liverpool School of Tropical Medicine
Pembroke Place
Liverpool
L5 3QA
Caitlin.thompson@lstmed.ac.uk
Manuscript ID: Spectrum05044-22

Dear Dr Christina Duomo, Editor In Chief

Thank you for allowing us to submit a revision of our manuscript entitled "*Multicentre diagnostic evaluation of OnSite COVID-19 Rapid Test (CTK Biotech) among symptomatic individuals in Brazil and The United Kingdom*". We express our gratitude for the valuable review of our paper, all the comments have helped us to further strengthen the overall quality of the paper and we have incorporated the suggestions as proposed in order to improve the manuscript.

Please find below our response to all comments. For clarity, reviewer comments are in black and ours are consistently in red. The revision has been developed in consultation with all co-authors, and each author has given approval to the final form of this revision.

Thank you for your consideration, please don't hesitate to contact me if you require any clarification on the edits made.

Sincerely,

Ms Caitlin Thompson, PhD candidate

Reviewer comments:

Reviewer #1 (Comments for the Author):

Major comments for authors

1. Methods: Line 112-114: "The mutation panel was customized to detect the variants: Alpha (P681H), Beta (E484k + K417N), Gamma (E484K + T20N), Delta and Kappa (L425R + P681R), Zeta (E484K) and Lambda (L452Q)." Using single mutation to differentiate different variants may be not reliable as different variants may share the same mutations, e.g., P681H shared by B.1.1.529 (Omicron) and B.1.1.7 (UK). Please provide references to justify this statement in 112-114.

Thank you for your suggestion.

In Brazil, the patients were enrolled from July to October 2021 before the emergence of the Omicron variant. The first Omicron cases in Brazil (and South America) were described on November 25, 2021 and the third wave of COVID-19 pandemic started at the end of December 2021 (1).

The TaqPath™ SARS-CoV-2 Mutation Panel Assay was customized to test different S-gene mutations (P681H, E484K, K417N, L452R, T20N, P681R and L452Q) to identify each variant. Later, the Omicron variant emerged with several mutations previously described in different VOCs as P681H, K417N, etc.

1. da Silva MS, Gularte JS, Filippi M, et al. Genomic and epidemiologic surveillance of SARS-CoV-2 in Southern Brazil and identification of a new Omicron-L452R sublineage. *Virus Res.* 2022;321:198907. doi:10.1016/j.virusres.2022.198907

The RT-qPCR genotyping method has been complemented to clarify this point as follows:

"For the detection of SARS-CoV-2 variants, samples were amplified using TaqPath™ 1-Step RT-qPCR Master SARS-CoV-2 Mutation Panel Assay (40X) (Thermo Fisher Scientific, Waltham, USA) following manufacturer instructions. RT-qPCR was prepared, and samples were tested for the presence of each S-gene mutation. The mutation panel was customized to detect each variant as follows: Alpha (P681H[+], E484K[-], K417N[-], L452R[-], T20N[-], P681R[-], L452Q[-]), Beta (E484K[+], K417N[+], P681H[-], L452R[-], T20N[-], P681R[-], L452Q[-]), Gamma (E484K[+], T20N[+], K417N[-], L452R[-], P681H[-], P681R[-], L452Q[-]), Delta and Kappa (L452R[+] P681R[+], E484K [-], K417N[-], T20N[-], P681H[-], L452Q[-]), Zeta (E484K[+], K417N[-], L452R[-], T20N[-], P681H[-], P681R[-], L452Q[-]) and Lambda (L452Q[+], E484K[-], K417N[-], P681H[-], L452R[-], T20N[-], P681R[-]) (2-4). Data was analyzed by QuantStudio™ design and analysis software v2.5.1. in the genotyping module. The variant was identified according to the positivity for each mutation tested" (Lines 110-121)

References N° 2, 3 and 4 were added as suggested in the method section:

2. Ashford F, Best A, Dunn SJ, et al. SARS-CoV-2 Testing in the Community: Testing Positive Samples with the TaqMan SARS-CoV-2 Mutation Panel To Find Variants in Real Time. *J Clin Microbiol.* 2022;60(4):e0240821. doi:10.1128/jcm.02408-21
3. Castro GM, Sicilia P, Bolzon ML, et al. Tracking SARS-CoV-2 Variants Using a Rapid Typification Strategy: A Key Tool for Early Detection and Spread Investigation of Omicron in Argentina. *Front Med (Lausanne).* 2022;9:851861. Published 2022 May 17. doi:10.3389/fmed.2022.851861
4. Neopane P, Nypaver J, Shrestha R, Beqaj SS. SARS-CoV-2 Variants Detection Using TaqMan SARS-CoV-2 Mutation Panel Molecular Genotyping Assays. *Infect Drug Resist.* 2021;14:4471-4479. Published 2021 Oct 27. doi:10.2147/IDR.S335583

2. Methods: Line 138-146: Analytical sensitivity for SARS-CoV-2 wild type (Pango, B1), Alpha (B.1.1.7), Delta (B.1.617.2), Gamma (P.1) and Omicron (B.1.1.529) were tested. It seems that Beta variant, B.1.351 (S. Africa) is missed. Please clarify.

Thank you for this comment. You are correct, this Beta variant, B.1.351 (S.Africa) was not included in the analytical evaluation of this test as this variant accounted for less than 0.01% of SARS-CoV-2 infections in the UK during July-August 2021 at the time of recruitment for this study.

1. Next Strain. *Genomic epidemiology of novel coronavirus - Global subsampling.* 2022; Available from: https://nextstrain.org/ncov/gisaid/global?f_country=United%20Kingdom
2. Our World Data. *SARS-CoV-2 variants in analyzed sequences, United Kingdom.* 2022 [cited 2022; Available from: <https://ourworldindata.org/grapher/covid-variants-area?country=~GBR>.

3.1. Table 1: multiple typos/errors were found as below:

Symptoms present for Brazil: 494/495=99.8% rather than 99.6% as indicated in the Table.

Gender for UK: "52.1% (110/210)" should be "52.4% (110/210)".

Vaccinated for UK: "132, 62%": should be 62.6% or 63% (132/211) depending on how many decimal places the author prefer. Definitely 62% indicated in the Table 1 is inaccurate.

Vaccination not disclosed (n, %) for UK : "1%" should be 0.47% (1/211)

Thank you for this feedback. Table 1 has been updated to correct these errors and to ensure all percentages are to 1 decimal place.

3.2. The total numbers for calculation of different indexes were different, e.g., 496 for age whereas 495 for gender for Brazil, 211 for age whereas 210 for gender for UK. Please explain.

Thank you for this observation. 496 patients were enrolled in Brazil and 211 patients were enrolled in the UK. We have one patient in Brazil and one patient in the UK who did not want disclosed their gender. This has been clarified on the Table footnotes.

3.3. There are different formats of data presentation in the Table 1, e.g., "7, 1%" and "6.5%, (32/496)".

Some percentages have one decimal place (e.g., 99.6%) while others have no decimals (e.g., 4%). The format in the Table 1 should be consistent among different rows.

Thank you for this observation, all data has now been amended to a format to a one decimal place percentage.

3.4. Please correct the corresponding parts in the main text.

Thank you for this feedback. All discrepancies with the data in the tables have been amended and corrected in the corresponding parts in the main text.

4.1. Table 2: invalid rate: 0.6% (3/496): what does "496" come from? There are a lot of 495 samples in Brazil, right? Please clarify.

Thank you for this observation. To clarify, there were 496 participants enrolled (also indicated in Table 1) Of these 496 participants, 495 had valid PCR results, therefore one participant was removed due to not having reference test results (PCR). The data was incorrectly reported for invalid rate and was 1/496, 0.2% rather than 3/496, 0.6%. This has been updated in the manuscript.

4.2. Table 2: the unexpected lower sensitivity of the OnSite test may be due to a limited number of PCR-confirmed positive samples enrolled in this study, e.g., 31 PCR positive samples in Brazil and 77 PCR positive samples in UK. The authors should test more PCR-confirmed positive samples by the OnSite test to further examine the clinical sensitivity.

We appreciate this observation and we acknowledge that this is a limitation of the study. However, the low number of positives reflected the low prevalence of COVID-19 at the time of recruitment in the UK and Brazil. We enrolled participants over a defined time period; in the UK between July and August 2021, in Brazil between July and October 2021. In future studies, we will ensure minimum positivity rates are required to examine clinical sensitivity.

5.1. Table 3: There are different formats of data presentation in the Table 3, e.g., "1, 3%" and "90.5%, 42".

Some percentages have one decimal place (e.g., 95.0%) while others have no decimals (e.g., 1%). The format in the Table 3 should be consistent among different rows.

Thank you for highlighting this discrepancy, the data in Table 3 has been updated to a one decimal place format.

5.2. Ct > 25 (n, %): 7, 22% should be 22.6% or 23% (7/31) depending on how many decimal places the author prefer. Definitely 22% indicated in the Table 3 is inaccurate.

Thank you for highlighting this error, the manuscript has been updated to the correct value and a one decimal place percentage in line with the rest of the data.

6. Statistical analysis: please clarify the statistical analysis were two-tailed or one-tailed.

Thank you for this comment. This should have been clarified in that the fishers exact and Chi Squared data analysis was two-tailed. This has been updated in the manuscript to reflect this clarification (LINE 162)

Minor comments for authors

7. Line 46 "clinical sensitivity than claimed by the manufacturer..." Please change "... to "."

Thank you for highlighting this error. This has been amended.

8. Line 58-60 "However, since April 2022 the UK government has ceased free Ag-RDT testing, now requiring the responsibility of the acquisition, and performance of tests to be placed on the individual." What does it mean by "now requiring the responsibility of the acquisition, and performance of tests to be placed on the individual"? Please clarify.

Thank you for highlighting this potential issue, as this sentence is not clear in its meaning, we have revised the wording to make this clearer for the reader to understand. This sentence now reads "However, since April 2022 the UK government has ceased free Ag-RDT testing, now requiring the responsibility of the purchase and use of the test to be placed on the individual." (LINE 62)

9. Line 123-125: "Swabs were taken systematically, NP swab samples in UTM (Copan Diagnostics Inc, Italy) were collected for the reference RT-qPCR test, this was followed by an NP swab to perform the Ag-RDTs." This sentence has grammatical issue so please re-write it. In addition, what does it mean by "swabs were taken systematically"? Please clarify.

Thank you for your suggestion, this sentence has been re-written to correct the grammatical issues. "Swabs were taken systematically; first an NP swab sample in UTM was collected from the patient for the reference RT-qPCR test, then an NP swab sample was taken to perform the Ag-RDTs." (LINES 130-132)

To clarify, the word "systematically" was used to describe the process of swab taken to clarify a process that is consistent and methodical.

10. Line 126-127: "All samples were transported in cooler boxes to the Liverpool School of Tropical Medicine (LSTM) and processed upon..." What is the shipment conditions and temperature in the cooler boxes?

Thank you for this comment, the samples were transported in cooler boxes with cool blocks for cold chain transport (+2C to +8C) during the transit to the laboratory (within 2h). The temperature in the cooler boxes was not monitored during the course of this study so we cannot comment on the exact temperature.

Reviewer #2 (Comments for the Author):

– 1-Give full names of all the abbreviations used first, then you can use abbreviations.

Thank you for this feedback. The manuscript has now been amended to include all full names before the use of abbreviations.

→ The English language of the article also needs to be improved

Thank you so much for your feedback, the manuscript has now been reviewed by a native English speaker and we have addressed the grammar mistakes and sentences that were unclear.

→ Italicize all the "et al., or et al." and scientific words (line 105).

Thank you for highlighting this error. This has been amended.

→ The discussion part needs a more detailed analysis of the results.

Thank you for this feedback, the discussion has been expanded to give a more detailed analysis of the results.

April 17, 2023

Dr. Ana I. Cubas-Atienzar
Liverpool School of Tropical Medicine
Liverpool
United Kingdom

Re: Spectrum05044-22R1 (Multicentre diagnostic evaluation of OnSite COVID-19 Rapid Test (CTK Biotech) among symptomatic individuals in Brazil and The United Kingdom)

Dear Dr. Ana I. Cubas-Atienzar:

Your manuscript has been accepted, and I am forwarding it to the ASM Journals Department for publication. You will be notified when your proofs are ready to be viewed.

Sincerely,

Rosemary She
Editor, Microbiology Spectrum
